# A New Type of Dynamic Vibration Fiber Sensor

**DOI:** 10.3390/s24216973

**Published:** 2024-10-30

**Authors:** I-Nan Chang, Chih-Chuan Chiu, Wen-Fung Liu

**Affiliations:** 1Department of Electronic Engineering, Feng Chia University, Taichung 40724, Taiwan; 2Department of Electrical Engineering, Feng Chia University, Taichung 40724, Taiwan; r11k43005@ntu.edu.tw (C.-C.C.); wfliu@fcu.edu.tw (W.-F.L.)

**Keywords:** fiber Bragg grating, vibration wave sensors, fiber sensors

## Abstract

A new-type vibration sensor based on a fiber Bragg grating combined with a special structure-packaged design is proposed for monitoring the mechanical vibration signals. Three different sensing structures, including the film squeeze type, new film squeeze type, and elastic tape squeeze type are proposed for measuring the vibration signals with the frequency range from tens to thousands of Hz. In the comparison to experimental results, the new film squeeze structure has a nice sensing performance in the range from 100 to 1000 Hz with a sensitivity of 0.302 mV/g. For the elastic tape squeeze structure, the elastic tape is designed to encapsulate the optical fiber with a good frequency response from 1100 to 3100 Hz. In addition, by using the new film squeeze structure to measure the steady-state and non-steady-state vibration signals, the spectral components of sensing signals are analyzed by using the wavelet transformation for confirming the testing signals. These vibration fiber sensors can be applied in the measurement of high-end manufacture-facility vibration or earthquake vibrations etc.

## 1. Introduction

In recent years, the development of fiber grating sensors has greatly attracted the attention of researchers due to a wide range of applications. In comparison with the conventional sensors, there are many advantages, including the simple structure, long life-time, high sensitivity, nice stability, etc. [1]. Even in special or harsh operating environments, such as high temperature, high electric field, radiation, and corrosive gases, the sensing performance can be effectively maintained. For the field of measuring the vibration signals created from an instrument or facility, there are many different sensing technologies including the conventional eddy current vibration wave sensors, piezoelectric vibration sensors, soft PZT nanofiber sensors, and fiber grating sensors, etc. [2]. Generally, the vibration wave can be divided into a damped and non-damped vibration wave. The real vibration mode is damped vibration, in which the frequency of vibration can be a single frequency or a mixture of frequencies and the vibration period is included in the periodic and aperiodic vibration according to the different vibration sources. Based on the nice characteristics for sensing vibration signals, a variety of FBG-based vibration or accelerometer sensors have been proposed [3,4,5,6,7,8,9,10]. By utilizing a cantilever beam and fiber gratings, the fiber Bragg grating is one of the most important vibration or acceleration sensors [3,4,5,6,7]. For these previous publications and according to the parameters of light modulation, optical vibration sensors can be divided into three categories: light intensity modulation, light phase modulation and optical wavelength modulation [8]. In general, the simple vibration or acceleration sensors are mostly of one- or two-dimensional design. For a one-dimensional FBG accelerometer or vibration meter, the FBG is typically attached to a mass-spring system. When the system senses acceleration or vibration signals, it causes a displacement of the mass, which induces a strain on the FBG to in turn cause a fiber grating wavelength shift. By monitoring the grating wavelength shift, the acceleration or vibration can be accurately obtained. By adjusting the effective distance between the sensor axis and the cantilever neutral axis, the sensor sensitivity can be effectively improved [9]. A string-type-based two-dimensional FBG vibration sensor is a nice technology that utilizes the transversal vibrational property of a tightly suspended optical fiber [10]. This type of sensor can be used in distributed two-dimensional vibration measurement due to its two-dimensional sensing properties. It is a significant advancement in the field of optical fiber sensing, where precision and innovation converge. In recent years, three-dimensional measurement has become increasingly important, so the design of three-dimensional sensors has gradually received attention [11]. The new three-dimensional acceleration sensor uses composite flexure hinges and fiber Bragg grating. The researchers investigated the coupling mechanism between a new integrated elastomer structure and fiber grating. They studied the influence of structural parameters on the static and dynamic characteristics, volume, and mass of the sensor.

Our purpose of measuring the vibration signals created from instruments or facilities is to monitor their operating performance for estimating the requirement of being maintained. For our proposed fiber grating sensors, the fiber grating combining the packaged design can be applied in checking the instrument or facility whether or not it is maintained according to the sensing spectrum.

## 2. Basic Sensing Principle

The key element of this vibration fiber sensor is the fiber Bragg grating, which is one of the potentially passive fiber components. For the fabrication of fiber gratings, the most general method is the phase-mask fabricating technique firstly proposed by K.O. Hill et al. in 1993 [12,13]. The ultraviolet light beam (248 nm) from a KrF excimer laser passes through the phase mask to form the interference fringe of ±1 order diffraction beams to exposure on the optical fiber to form the fiber grating with a periodical index variation along the fiber core axis. When the measuring light source from a broad-band light source is launched into a fiber grating, the forward fundamental propagation mode is coupled to the backward fundamental core mode to cause reflection light with a specific wavelength, called the grating Bragg wavelength (λ_B_), and the relationship between the Bragg wavelength and the grating period is shown in Equation (1):(1)$$\lambda_{B}=2n_{eff}\Lambda$$
in which n_eff_ is the effective index of grating and Λ is the grating period. From the formula, we can see that the different fiber-grating wavelengths can be easily fabricated by changing the grating period (i.e., the period of phase mask). For writing high reflectivity fiber gratings, the general commercial single mode fiber (SMF-28, Corning, New York, NY, USA) requires hydrogen-loading for increasing the fiber photosensitivity. The operating mechanism for measuring the vibration signal by using a fiber grating is to combine the design of fiber-grating packaged-structure and to induce the strain caused by the vibration signals to create the grating center-wavelength shift.

For the design of the sensor structure, since the natural resonance frequency of the structure itself is fixed, if the resonant cavity structure is fabricated to amplify the signal amplitude, it may be limited by the vibration wave of a specific frequency or due to the compressibility of the air to cause the vibration wave to be absorbed. Therefore, the design of the film squeeze structure mainly refers to the traditional load-bearing extrusion method of piezoelectric sensors. During the vibration process, the piezoelectric material is deformed by the squeeze of the weight and the process of generating electrical signals. The sensing mechanism of these proposed fiber sensors is based on the grating wavelength shift caused by the vibration signals from the speaker. The speaker acts as a vibration signal source to create the vibration signals to drive the elastic silicon thin-film to result in the spontaneous vibration of the thin-film for stretching and depressing the sensing fiber grating to cause the grating wavelength shift. The configuration of the operating mechanism for this sensor is shown in Figure 1. This idea is to combine the fiber grating with the special structure-packaged design. Firstly, the fiber grating is embedded in a silicone rubber film with a thickness of 277 μm, and then the film is cut into strips along the axial direction of the grating. The silicone rubber film is made by utilizing a mixture of SL-2500A (agents A and B) (Chih Yi Bussiness Company, New Taipei, Taiwan). For each batch, agents A and B are mixed in a 10:1 ratio. The prepared solution is poured into the center of a circular iron plate with a diameter of 10 cm and placed in a spin coater. The spin coater uses centrifugal force to evenly distribute the silicone rubber solution across the iron plate. By adjusting the amount of silicone rubber solution or controlling the speed and duration of the spinning process, films of different thicknesses can be produced. A 3D printing technique is used to print a circle that is closed on one side. One end of the fiber grating is passed through the circular film, and finally, the whole is fixed to the cylindrical structure using silicone adhesive (3M Silicone Adhesive PN08019, 3M, Maplewood, MN, USA). In this way, the bottom film can be aligned with the vibration source, and through the upper and lower resonance effect of the bottom film. The central vertical film will be stretched and compressed to cause the grating to be strained and then to result in the grating center wavelength shift. By measuring the grating wavelength shift, the vibration signals can be obtained. From the experiments, the elastic tape (3M Polyester Tape 8901, 3M, Maplewood, MN, USA) has nice tensile and elastic effects for better frequency response. The film fiber grating is fabricated by a fiber grating embedded with an elastic film to compare the sensing performance between the two different sensors.

In addition, the 3D printing time of the film squeeze structure is longer and, during the measurement process, the sensing height must be fixed by hand-holding or using the object to stand up to cause the structure deformation or to create additional protruding blocks. For solving the problem, the support frame is designed to stabilize the overall structure for measuring the vibration signals according to the appropriate fixed height. The 3D view and cross-section of the new film squeeze structure are shown in Figure 2.

The operating mechanism of sensors is based on that the film of sensors can sense vibrational signals to transfer the vibrational force to squeeze and extrude the fiber grating to cause the grating wavelength shift. When a uniform stress (P) is applied in the sensing fiber grating along the axial direction, it will create a fiber strain (ε) in the fiber grating, ε = (υP)/E, where υ is Poisson’s ratio and E is the fiber module parameter. The strain in a fiber grating will cause the variation both of grating period and grating effective index and then to create the grating wavelength shift. This equation can be given as the following
(2)$$∆\lambda_{B}=2\Lambda ∆n_{eff}+2∆\Lambda n_{eff}$$
where ∆Λ is the grating period variation induced by the strain-elastic coefficient and ∆n_eff_ is the effective index variation induced by the strain-optic effect.

Equation (2) can be further indicated as
(3)$$\Delta \lambda_{B}=2\Lambda \left(\frac{\partial n_{eff}}{\partial L}\Delta L+\frac{\partial n_{eff}}{\partial d}\Delta d\right)+2\frac{\partial \Lambda }{\partial L}\Delta Ln_{eff}$$
where ΔL is the grating physical length variation along the fiber axis and ∆d is the fiber diameter variation in the radial direction. $\frac{\partial n_{eff}}{\partial L}$ is the strain-optic effect and $\frac{\partial n_{eff}}{\partial d}$ is the photo-waveguide effect which can be neglected due to being very small value. By using Taylor-series expansion to neglect the high order terms and to introduce the strain-optic coefficient components of fused-silica (p_ij_), Equation (3) can be given as
(4)$$\frac{\Delta \lambda_{B}}{\lambda_{B}}=\left\{1-\frac{{n^{2}}_{eff}}{2}\left[p_{12}-\left(p_{11}+p_{12}\right)v\right]\right\}\varepsilon_{zz}=S_{e}\varepsilon$$
where S_e_ is the strain-induced coefficient of the relative grating wavelength shift and υ is the Poisson ratio.
(5)$$S_{e}=1-\frac{{n^{2}}_{eff}}{2}\left[p_{12}-\left(p_{11}+p_{12}\right)v\right]$$

## 3. Experimental Setup and Results

For the experiment, the source of vibration signals is a full-range speaker with 8 Ω-3 W. Each sensor is fixed at a support frame to have a height of 45 mm above the speaker. The signal generator generates a sine wave signal with an amplitude of 300 mV_pp_ to be amplified by a signal amplifier to create 6 V_pp_ signals to drive the speaker as shown in Figure 3. The output frequency of the signal generator is tuned in the range from 20 to 2000 Hz with a 20 Hz interval for the overall measurement. The frequency of vibration signals from 0 to 100 Hz is defined as the low-frequency band, 100 to 1000 Hz is defined the mid-frequency band, and above 1000 Hz is defined as the high-frequency band.

For confirming the frequency response of the output signals of the speaker, we firstly use the no-packaged bare-fiber grating to directly glue on the surface of the speaker to sense the vibration signals. This original frequency response can be obtained for providing a comparison with that of another sensing heads. The original frequency response is obtained, as shown in Figure 4, acting as a benchmark for comparing the sensing performance by using different packaged-structures.

The overall experimental set-up is shown in Figure 5, in which the output light of the broadband spontaneous emission light source (ASE Light Source) is put into the first fiber circulator and then launched into the matching fiber grating (Match Gating, MG) to obtain the grating reflection signal, which is input into the second fiber circulator. The overlapping spectrum between the matching grating and sensing grating (SG) from port 3 of the second fiber circulator is sent to the photodetector (InGaAs), and finally converted into an electrical signal by the photodetector and connected to an oscilloscope. In addition, the center wavelength of the matching fiber grating and the sensing fiber grating must be the same, because the light energy finally transmitted to the photoelectric converter will be determined according to the overlapping area of the sensing center wavelength and the matching center wavelength. The demodulation mechanism of grating wavelength shift based on the overlapping reflection spectra between the MG and the SG is shown in Figure 6. During the process, the change in light energy will be displayed on the oscilloscope, and the waveform will be reflected as a sine wave signal, so that the amplitude and frequency of the vibration wave can be observed and analyzed.

For comparing the experimental results in different sensing structures, the film squeeze type is used for detecting vibration signals in the mid–high frequency range from 260 to 1000 Hz and in the mid–low frequency range from 100 to 150 Hz. Because the vibration amplitude of the bottom film is too large, the irregular oscillation makes the waveform distortion as shown in Figure 7a. In the intermediate frequency band, although the amplitude is not enough large, the waveform can be measured stably as shown in Figure 7b,c. For the signals above 2000 Hz, it is difficult to measure, due to the small vibration amplitude and the soft absorption characteristics of the film. The overall signal measurement with its frequency response spectrum is shown in Figure 8.

The elastic-tape squeeze type shows its material properties for the frequency range from 100 to 200 Hz, and has a high signal amplification effect, but also due to the excessive deformation of the fiber grating, the waveform distortion is shown in Figure 9a,b. However, the characteristic of this structure is that it has a significant sensing effect in the high-frequency range from 1100 to 2500 Hz. The main reason is that the stretching effect of the elastic tape is better, and it is not easy to absorb the signal amplitude, so that it can measure up to about 3100 Hz. The sensor with better performance in the mid- and high-frequency response is shown in Figure 9c,d. We can also observe that in comparison with unstructured fiber grating, the high-frequency response is nice than that of a pure grating as shown in Figure 10. The sensor with the best high-frequency response in the experiment is an elastic tape squeeze structure.

For improving the low-frequency sensing performance, a probe is installed at the bottom of the elastic tape squeeze sensor, as shown in Figure 11, by directly contacting the vibration source created by the speaker. The purpose of the probe structure design is to obtain a better response in the lower-frequency band by utilizing the probe tip to directly contact the vibration film of speaker to directly conduct the speaker vibration signal to drive the silicon thin-film. The purpose of this design is to compare the sensing characteristics between the non-contact type and contact type fiber-grating vibration sensors. The probe is wooden material with a length of 3.6 cm, a width of 3.6 cm, and a height of 4.2 cm. This design is attributed to the fact that the sensing thin-film of the sensor can directly detect the larger vibration signals in the lower frequency band created from the speaker. It is confirmed by the experimental results, as shown in Figure 12, which show the capability to detect the 10 Hz signal. Due to the small vibration amplitude in the middle- and high-frequency bands, the sensing performance will be decayed by using the probe contact method.

For solving the limitation of the film squeeze sensor that can only be held at a fixed height, a new film squeeze structure is proposed to improve this problem. The frequency response spectrum of the new film squeeze structure is shown in Figure 13. We can see that, owing to the effect of the sensing support fixing frame, the amplitude gain response is nice in the frequency from 100 to 250 Hz, but the film vibrates with too large an amplitude to cause the drift of the grating center wavelength to exceed the matching grating range and then to sense the waveform with the phenomenon of positive peak interception, as shown in Figure 14a. This is attributed to the high sensitivity of the new film structure sensor in this frequency band. If the amplitude of vibration signal is decreased, this sensor has a nice capability to measure the signal of this frequency band as shown in Figure 14b. This sensor has a stable sensing performance in the mid-frequency range from 100 to 1000 Hz, as shown in Figure 14c,d.

However, in order to clearly understand the sensing sensitivity of each sensor, a sine wave from the signal generator with a frequency of 300 Hz is amplified and adjusted to be the output voltage range from 2 V_pp_ to 20 V_pp_ per 2 V_pp_ to drive the speaker. The output signals of the photodetector detecting from the sensing-head optical signals are monitored for obtaining the relationship between the output and input amplitudes, as shown in Figure 15. From the figure, we can see that the measurement curves between the output and the input voltages for the sensors are almost linear. The sensitivity of each sensor is the curve slope, which can also be linearly fitted from each line segment in the figure. From the curves, a sensitivity of 35 mV/V, which can be converted to 0.302 mV/g according to the speaker specifications, the driving speaker voltages, and detecting signals, etc. The new film squeeze type at 300 Hz is obtained and the sensitivity of the elastic tape squeeze type is 15 mV/V, which can be converted to 0.129 mV/g according to the speaker specifications, the driving speaker voltages, and detecting signals, etc., at 300 Hz. Thus, the sensitivity of the new film squeeze structure in the intermediate frequency is higher than that of other types of sensors.

In addition to monitoring the time-domain signals from the output signals of the photodetector, it is required to obtain frequency information for the sensing signals. Different digital signal algorithms are used for confirming the frequency spectrum of sensing signals. The actual sensing signals are usually not a single frequency or a stable amplitude. By using the Fourier transform, it is difficult to simultaneously obtain the signal spectrum at various time points. Therefore, in order to improve this problem, the wavelet analysis (Continuous Wavelet Transform, CWT) is used for this signal analysis [14]. The following formula shows the theoretical expression of wavelet analysis:(6)$$W\left(a,b\right)=\int_{-\infty }^{\infty }x\left(t\right)\frac{1}{\sqrt{a}}\bar{\psi }\left(\frac{t-b}{a}\right)dt,$$
where *ψ*(*t*) is a continuous function in both time domain and frequency domain called the mother wavelet and the over-line represents operation of complex conjugate. *a* is the scaling parameter, *b* is the translation parameter, and the factor (1/√*a*) is used to normalize the energy of the CWT. The main purpose of the mother wavelet is to provide a source function to generate the daughter wavelets, which are simply the translated and scaled versions of the mother wavelet. Since it introduces a special window function for calculation, the low-frequency component is calculated with long-term series and narrow bandwidth and the high-frequency component is calculated with a short time series and a wide bandwidth to obtain the timing of the frequency to be reflected in the time series of the signal, as shown in Figure 16, in which the time-domain signal is measured by a new film squeeze sensor. By using the fast Fourier transform analysis, the vibration signal has frequency changes from 100 Hz to 300 Hz within 0.5 s, as shown in Figure 17, in which the peak of around 150 Hz can be used for confirming to match the peak of frequency response in Figure 13. From the comparison between Figure 13 and Figure 17, we can see that the frequency response of the sensor is mainly focused on the frequency range from 100 Hz to 300 Hz, but the frequency response obtained by using the FFT has a large amount of noise. It shows the sensing signals in components of 100~300 Hz. Figure 18 is a wavelet analysis diagram in which the color contrast is used to distinguish different time points and the amplitude of the components that appear at each frequency. The marked-red region shows that, within the time variation, the appearance time of frequency components from 100 Hz to 300 Hz extends from the bottom left to the top right. It can be known that the vibration signals whose frequency changes with time rather than a mixing signal can reduce the misjudgment of the spectrum.

## 4. Conclusions

In this paper, various packaged-structure fiber-grating vibration sensors are proposed. From the experimental results, we can see that the contact-type structure in the low-frequency vibration band has better sensing performance. The film type is suitable in the medium-frequency vibration wave. For high-frequency vibration waves, the sensing materials with more elasticity and tension are required due to the fact that the high-frequency extrusion and stretching effects can effectively cause the deformation of the fiber grating. In addition, in terms of signal analysis, a wavelet transform can be used simultaneously to analyze time-frequency signals for obtaining more time-frequency information. Thus, the sensor based on fiber gratings can effectively achieve the measurement of vibration signals created from high-end machines or facilities owing to the characteristics of high sensitivity, anti-corrosion, anti-electromagnetic interference, etc.

## Figures and Tables

**Figure 1 sensors-24-06973-f001:**
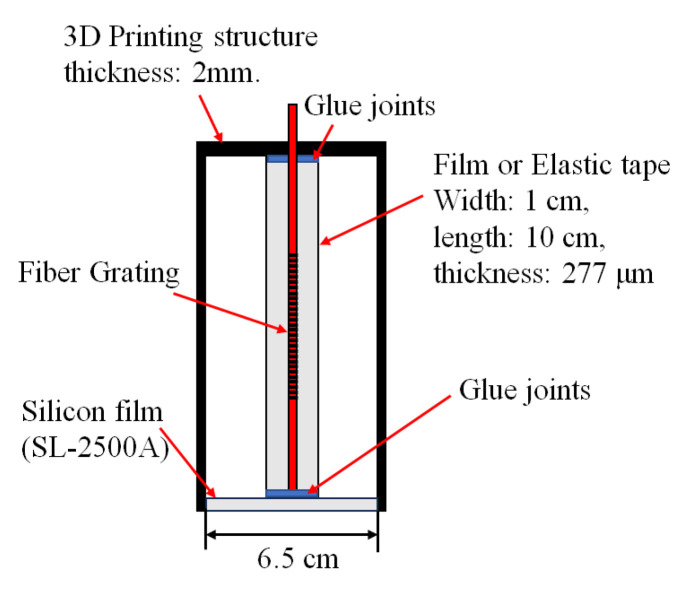
The cross-section of the package structure without the excitation of vibration signals.

**Figure 2 sensors-24-06973-f002:**
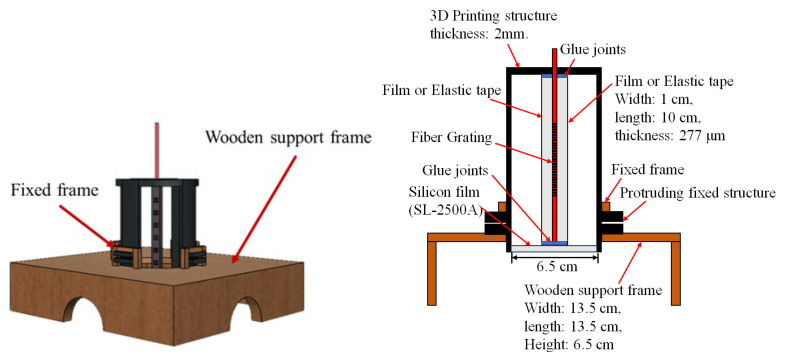
The 3D view and cross-section of the new film squeeze structure.

**Figure 3 sensors-24-06973-f003:**
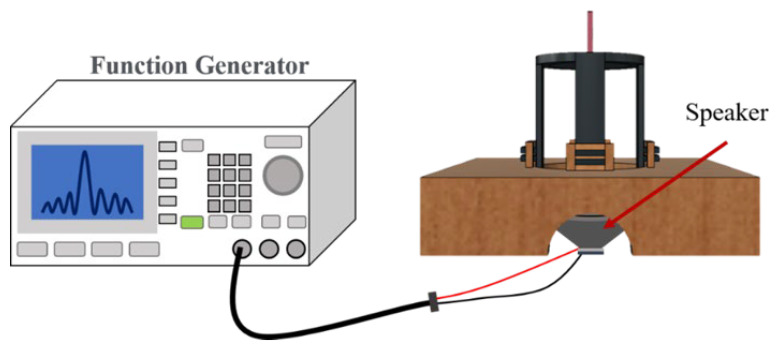
Experimental configuration of generating vibration signals.

**Figure 4 sensors-24-06973-f004:**
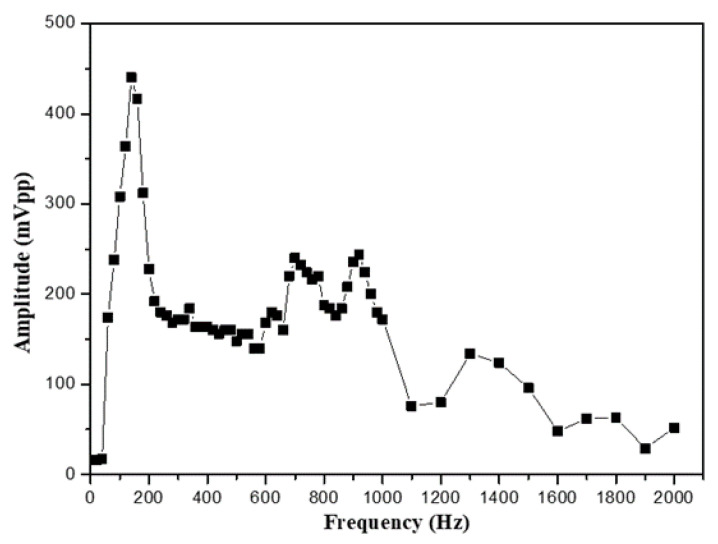
Frequency response of bare fiber grating for sensing vibrational signals.

**Figure 5 sensors-24-06973-f005:**
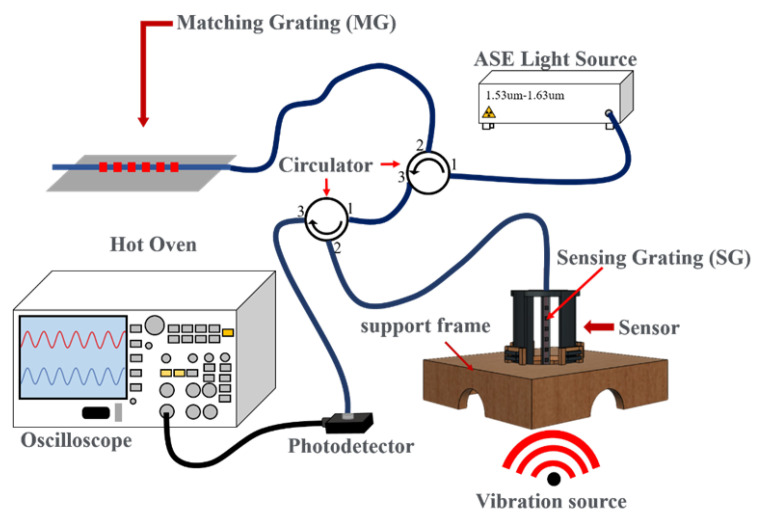
Experimental set-up of sensing vibrational signals.

**Figure 6 sensors-24-06973-f006:**
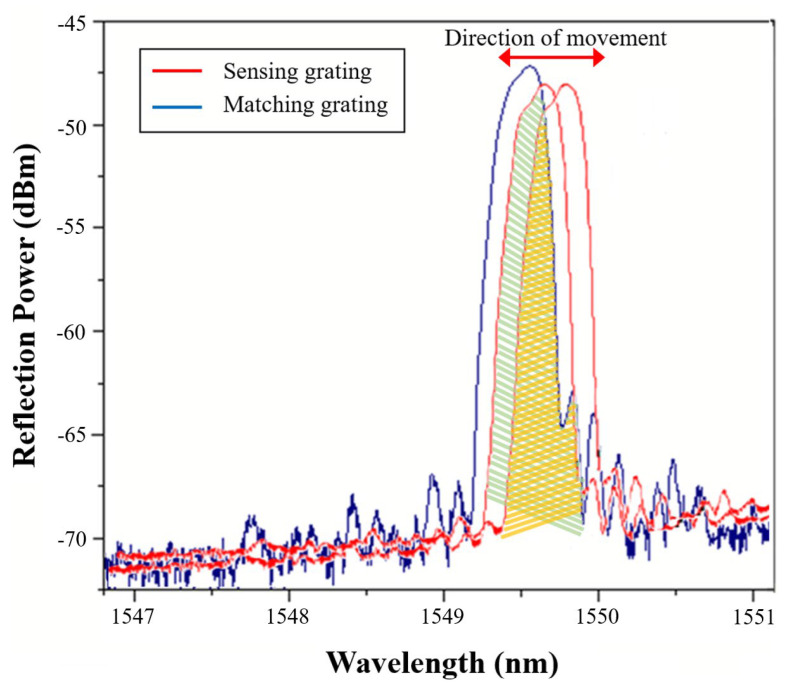
The overlapping reflection spectra between the sensing grating and the matched grating.

**Figure 7 sensors-24-06973-f007:**
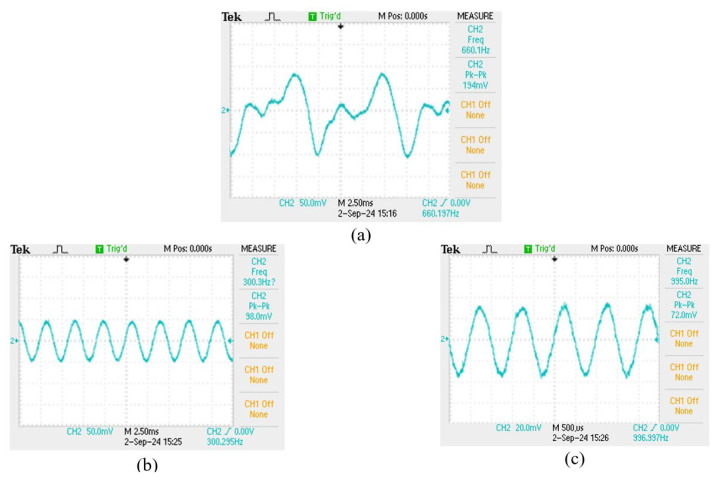
(**a**) 100 Hz sensing signal, (**b**) 300 Hz sensing signal, (**c**) 1000 Hz sensing signal.

**Figure 8 sensors-24-06973-f008:**
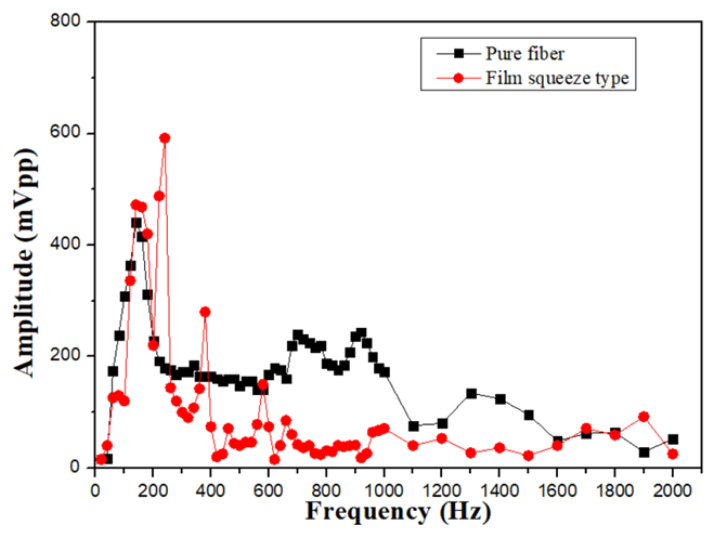
Comparison of frequency response of film squeeze type and bare fiber grating.

**Figure 9 sensors-24-06973-f009:**
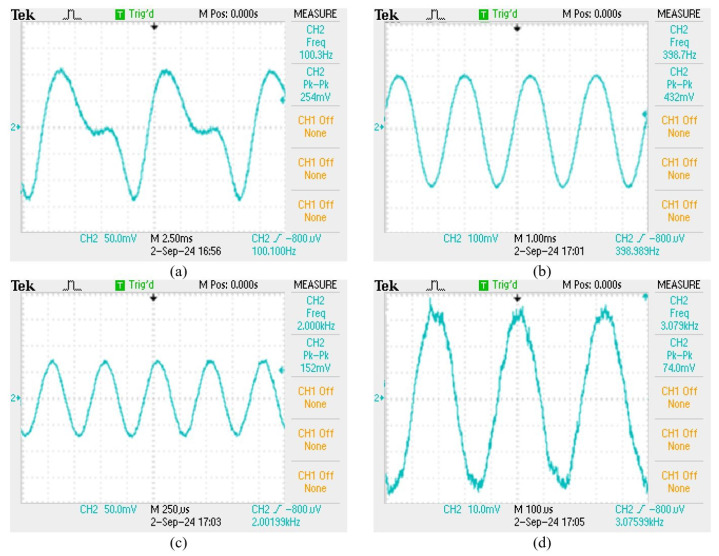
(**a**) 100 Hz sensing signal, (**b**) 400 Hz sensing signal, (**c**) 2000 Hz sensing signal, (**d**) 3100 Hz sensing signal.

**Figure 10 sensors-24-06973-f010:**
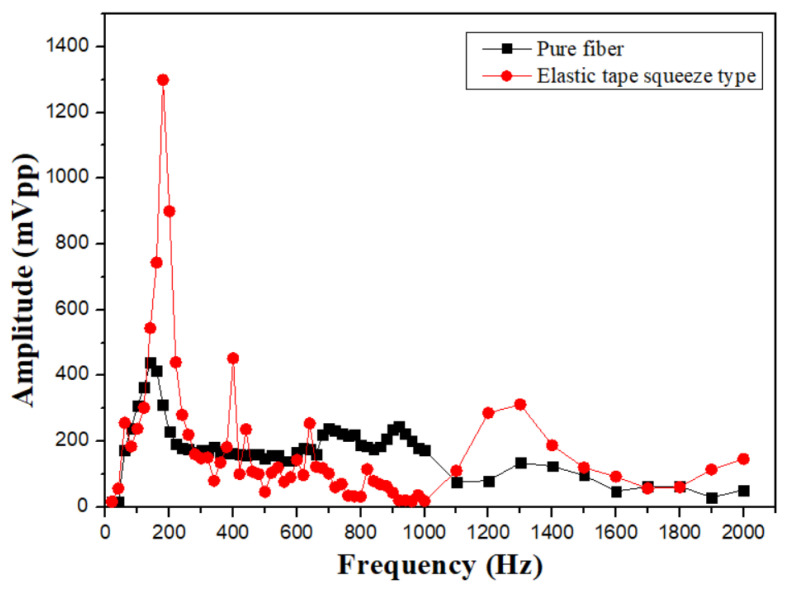
Comparison of frequency response of elastic tape squeeze structure and unstructured fiber grating.

**Figure 11 sensors-24-06973-f011:**
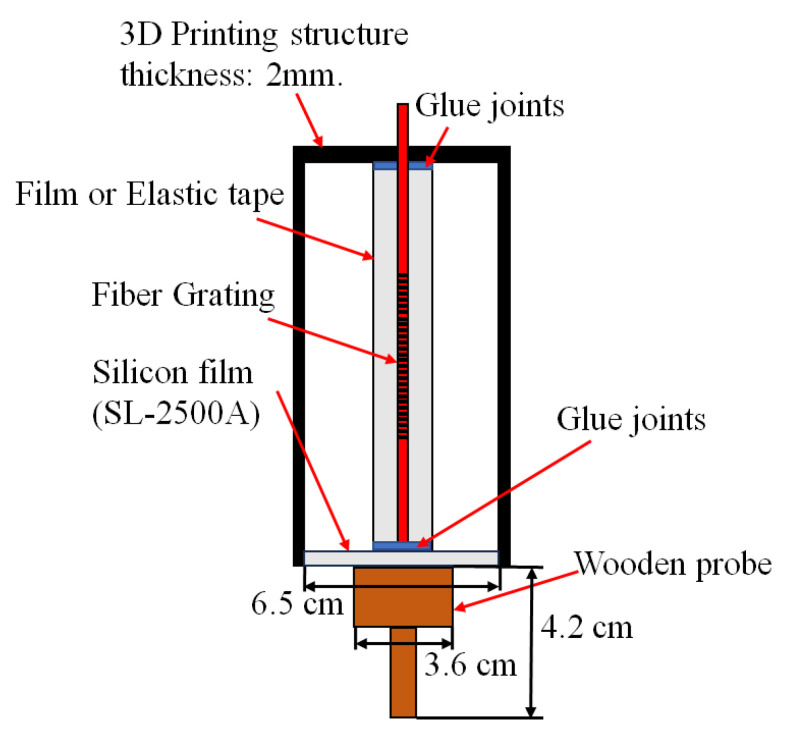
The cross-section of elastic-tape squeeze type with a probe.

**Figure 12 sensors-24-06973-f012:**
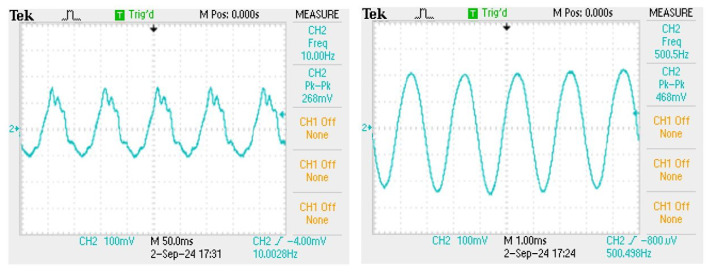
The sensing signal of 10 Hz and sensing signal of 500 Hz.

**Figure 13 sensors-24-06973-f013:**
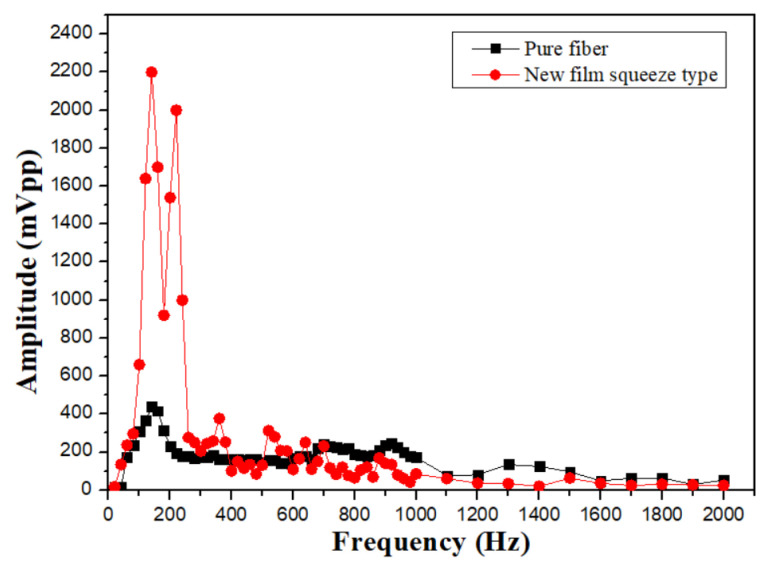
Comparison of frequency response of new film squeeze and unstructured fiber grating.

**Figure 14 sensors-24-06973-f014:**
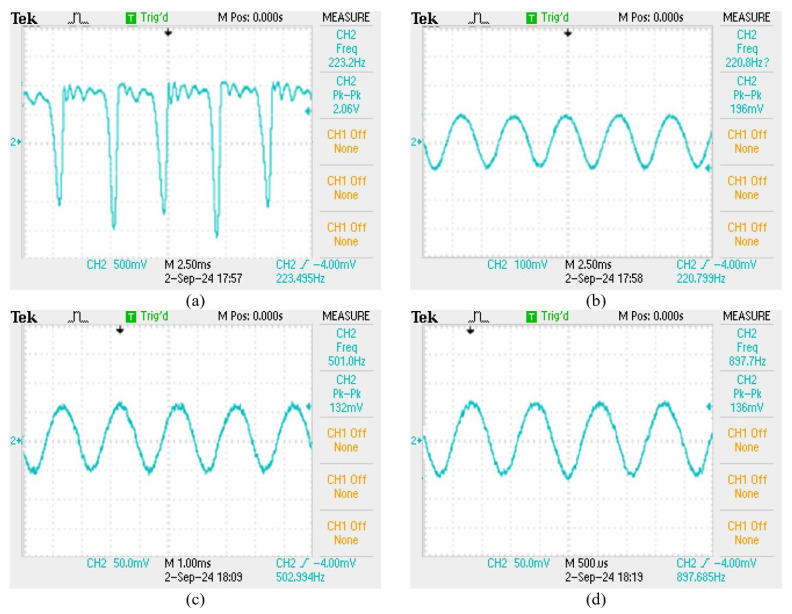
The sensing signals with different vibration amplitude and frequency. (**a**) 220 Hz sensing signal vs 300 mV vibration amplitude, (**b**) 220 Hz sensing signal vs 100 mV vibration amplitude, (**c**) 500 Hz sensing signal vs 300 mV vibration amplitude, (**d**) 900 Hz sensing signal vs 300 mV vibration amplitude.

**Figure 15 sensors-24-06973-f015:**
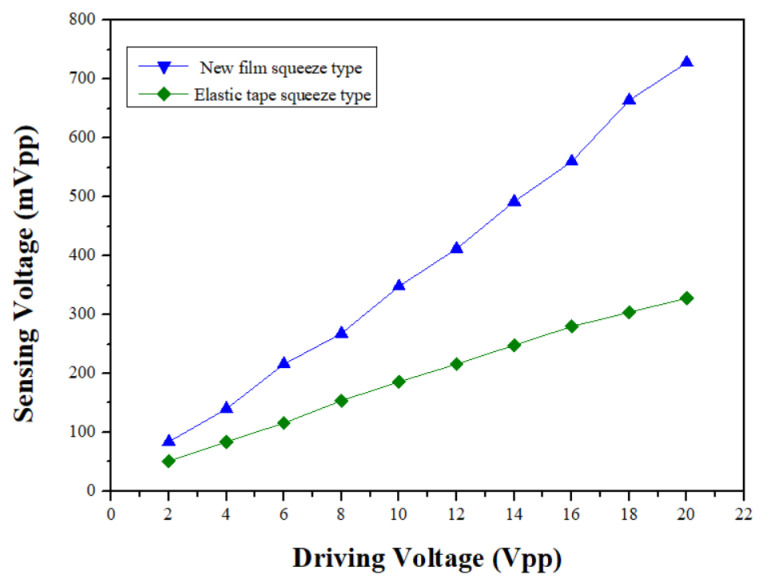
The relationship between vibration signal and sensing amplitude.

**Figure 16 sensors-24-06973-f016:**
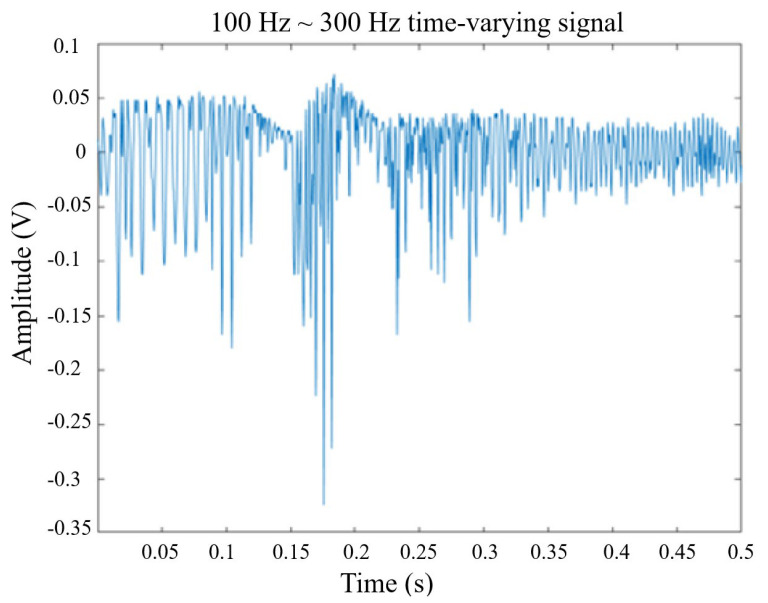
A signal frequency varies with time series (100~300 Hz).

**Figure 17 sensors-24-06973-f017:**
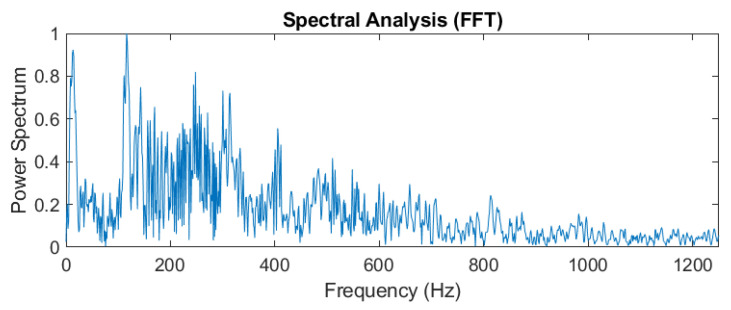
The time-frequency analysis of vibration signals with FFT.

**Figure 18 sensors-24-06973-f018:**
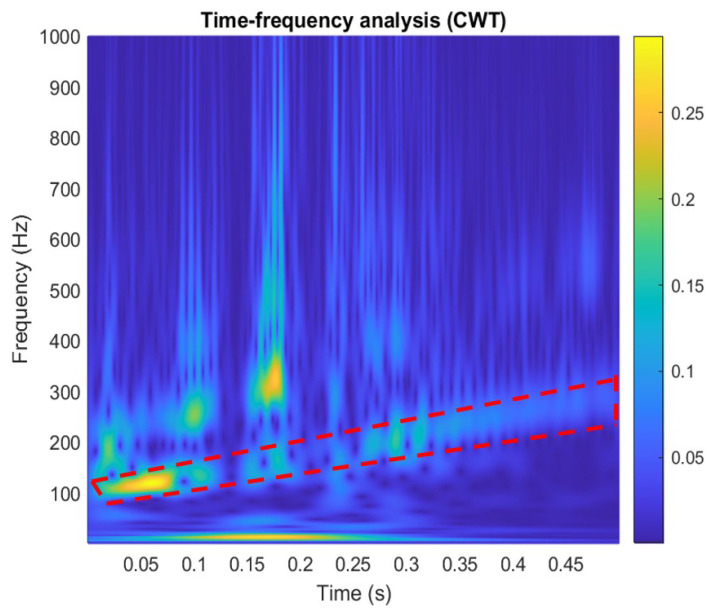
The time-frequency analysis of vibration signals with CWT.

## Data Availability

Data are contained within the article.

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
