# Peer review of "A New Type of Dynamic Vibration Fiber Sensor"

_sensors, 2024, doi:10.3390/s24216973_

Round 1
Reviewer 1 Report
Comments and Suggestions for Authors
1. Figure 16 shows that the peak of the noise spectrum is around 150 Hz. I am unsure whether this is a characteristic of the noise itself or a function of the new sensor. For certainty, it would be beneficial to include some mechanical frequency characteristics of the newly designed pump film structure. This means conducting an additional measurement or at least providing an estimation of the eigenfrequencies.
2. After Equation (16), the authors should clarify which function $\psi$ they refer to in their wavelet analysis.
3. The slope of the curve in Figure 14 is given in mV/V, meaning that it is normalized by the speaker’s voltage. For potential practical uses of the novel sensor, readers want to understand its sensitivity in more conventional units. Can the authors provide an estimate of the sensitivity using standard technical units?

Reviewer 2 Report
Comments and Suggestions for Authors
This work closely resembles a laboratory experiment for undergraduate students. At the very least, they should measure the noise generated by the optical fiber, which is more important than the vibration signal itself. Besides, any fiber can be used to measure the vibrations. What is new in that? I see there is no novelty, and the idea is very shallow to publish.
Author Response
Thanks for your comment.

Reviewer 3 Report
Comments and Suggestions for Authors
see attachment

Minor editing of English language required.
Round 2
Reviewer 2 Report
Comments and Suggestions for Authors
The paper underwent a thorough assessment, demonstrating its potential for publication.
Author Response
Thanks for the comment.
Reviewer 3 Report
Comments and Suggestions for Authors
Although the author has made careful revisions to the manuscript, the following issues still exist.
1. Although a schematic diagram Figure 1 (a) of a cross-sectional structure of the sensor was added in the revised manuscript, schematic diagrams of other cross-sections of the sensor structure should also be added to reflect more structural details and highlight its differences from other similar sensors. Additionally, Figure 1 (b) should be deleted as it does not reflect any details of the sensor structure.
2. Figure 2 and Figure 3 and Figure 11 are improvements to Figure 1, their structural details should be the most distinctive and creative aspect of the sensor. However, these figures are too blurry to reflect their uniqueness and innovation, and should be redrawn as cross-sectional diagrams as shown in Figure 1.
3. In addition to adding a cross-sectional view reflecting the details of the sensor structure as shown in Figure 1, it is also necessary to provide specific dimensional and mechanical parameters of the sensor structure, including 3D printed structure, elastic tape, silicon film and glue joints. Based on these dimensional and mechanical parameters, the digital simulation of sensor vibration characteristics should be complete to optimize the characteristic parameters of the sensor. If these dimensional and mechanical parameters are not carefully optimized, the illustrated structure may actually lead to a significant decrease in sensor characteristics.
4. The process details of elastic tape, silicon film and glue joints installation are also important for the characteristics of the sensor. However, there is no text or illustrations about them in the manuscript.
5. In the "2 Basic Sensing Principles", the author derived the theoretical equations (5) and (6) for vibration sensing based on the structural schematic diagrams in Figures 1, 2, and 3. But the formulas only contain strain without vibration, so they are force sensing formulas rather than vibration sensing formulas.
6. In vibration experiments, an excitation table must be used as the vibration source, and high-precision acceleration sensors must be used to calibrate the vibration values. But the author mistakenly used the sound experimental setup in Figure 4 in this vibration experiment. He confused vibration and strain in the theoretical analysis of the previous chapter, while confused vibration and sound in the experiments of this chapter.
7. According to formula 6, the strain sensed by the sensor will cause a change in the wavelength output of the sensing FBG, which is then converted into a change in light intensity through the combination of matching FBG and photodetector shown in Figure 6. When explaining the wavelength demodulation principle in Figure 6, it is recommended to use the spectral matching relationship diagram of two FBGs to explain the conversion relationship between wavelength and light intensity, rather than purely textual explanation.
8. In Figure 11, the author installed a probe at the bottom of the at the elastic tape squeeze sensor. However, the author did not specify what kind of probe and its readout instrument were used, nor did they explain the specific details of installing the probe at the bottom of the elastic tape squeeze sensor. Therefore, it is impossible to confirm the rationality of this experimental plan.
9. According to the experimental results in Figures 12 ~ Figures18, it can only be concluded that the sensitivity of the sensor is nv/V. But the author baselessly converted it to mv/g, which is unacceptable.
